

# Neural processing of working memory in adults with ADHD in a visuospatial change detection task with distractors

Chao Gu[1], Zhong-Xu Liu[2,3], Rosemary Tannock[2,4] and Steven Woltering[1,2]

[1] Department of Educational Psychology, Texas A&M University, College Station, TX, United States of America
[2] Department of Applied Psychology and Human Development, University of Toronto, Toronto, Canada
[3] Rotman Research Institute of Baycrest Centre, Toronto, Canada
[4] Neuroscience and Mental Health Research Program, Sickkids Hospital, Toronto, Canada

## ABSTRACT

Individuals with Attention-Deficit Hyperactivity Disorder (ADHD) are often characterized by deficits in working memory (WM), which manifest in academic, professional, and mental health difficulties. To better understand the underlying mechanisms of these presumed WM deficits, we compared adults with ADHD to their peers on behavioral and neural indices of WM. We used a visuospatial change detection task with distractors which was designed to assess the brain's ability to effectively filter out distractors from WM, in addition to testing for effects of WM load. Twenty-seven unmedicated adults with ADHD were compared to 27 matched peers on event-related potential (ERP) measures of WM, i.e., the contralateral delay activity (CDA). Despite severe impairments in everyday life functioning, findings showed no difference in deficits in behavioral tests of working memory for adults with ADHD compared to their peers. Interestingly, there were differences in neural activity between individuals with ADHD and their peers showing that the CDA of individuals with ADHD did not distinguish between high, distractor, and low memory load conditions. These data suggest, in the face of comparable behavioral performance, a difference in neural processing efficiency, wherein the brains of individuals with ADHD may not be as selective in the allocation of neural resources to perform a WM task.

## INTRODUCTION

Attention-Deficit/Hyperactivity Disorder (ADHD) is a prevalent mental health disorder characterized by persistent behavioral symptoms of inattention and hyperactivity/impulsivity, which can severely impede educational, occupational and health outcomes (*De Graaf et al., 2008*; *Polanczyk & Jensen, 2008*; *Uchida et al., 2018*). Visuospatial Working Memory (WM) impairments, as a sub-component of working memory (*Baddeley, 2010*), are impaired in individuals with ADHD compared to the general population, and can contribute to, or exaggerate, overall functional impairments (*Hervey, Epstein & Curry, 2004*; *Martinussen et al., 2005*; *Chamberlain et al., 2011*; *Fried et al., 2016*).

Corresponding author
Steven Woltering, swolte@gmail.com

WM refers to a limited-capacity, multi-component cognitive system that is required to keep relevant information in mind for a few seconds while performing complex tasks such as reasoning and comprehension (*Baddeley, 2010*; *Miyake & Shah, 1999*). Evidence of poor WM performance has generally been assumed to reflect a smaller capacity to store information, i.e., individuals with poor WM hold fewer items in mind compared to those with good WM (*Bunge et al., 2000*; *Cowan, 2001*). However, researchers have pointed to attention as a control process, which regulates access to WM, to be influential as well (*Awh, Vogel & Oh, 2006*; *Cowan et al., 2005*; *Shipstead et al., 2014*). In fact, this line of research suggests that storage capacity may be an outcome rather than a mechanism of WM, and that filtering efficiency (the ability to inhibit irrelevant information from entering into WM) may be the primary neurocognitive mechanism that helps control it (*Vogel, McCollough & Machizawa, 2005*; *Vogel & Machizawa, 2004*; *Kuo, Stokes & Nobre, 2012*).

In this study, we examined neural indices of storage capacity and filtering efficiency of adults with ADHD and their peers using a visuospatial change detection working memory task while electroencephalography (EEG) was recorded. Similar to *Vogel & Machizawa (2004)*, we measured the contralateral delay activity (CDA) which is an event-related potential (ERP; averaged EEG waves locked to an event) measured at posterior sites during the retention period. The CDA's amplitude increases reliably with the number of items to be remembered in the task's different memory load conditions; and has been shown to reach a limit at an individual's WM capacity (*Luria et al., 2016*). The difference in amplitude between low and high load conditions has been found to predict behavioral (i.e., neuropsychological) measures of storage capacity (*Vogel, McCollough & Machizawa, 2005*; *McCollough, Machizawa & Vogel, 2007*; *Diamantopoulou et al., 2011*; *Kundu et al., 2013*). We will refer to this measure as CDA-Δ.

The sensitivity of the CDA waveform to load can also be used to determine an index of filtering efficiency: in other words, whether irrelevant items unnecessarily consume WM capacity. In the distractor version of this task, participants are asked to remember the colors of squares (two squares in the low load and four in high load) in an array but to ignore the circles in a distractor condition (two squares + two circles). If the CDA of the distractor condition (two circles + two squares) resembles the low load condition (two squares), we can conclude that participants have effectively filtered out the irrelevant circles and excluded them from storage. Conversely, if the CDA for the distractor condition resembles that of the high load condition (4 squares), participants may have failed to efficiently filter out the distractors. *Vogel & Machizawa (2004)* and *Vogel, McCollough & Machizawa (2005)* have shown that the distractor CDA in people with low WM performance resembled their high load CDA and that, conversely, the distractor condition was similar to the low load condition in people with a high WM performance. More recently, studies have even found that training in visual filtering efficiency improves working memory capacity (*Li et al., 2017*). These findings suggest that people's WM performance is also strongly related to their filtering efficiency, i.e., their brains ability to filter out irrelevant items.

Reviews of the neural origins of the CDA have concluded that the waveform is likely generated by multiple sources in the brain, of which the posterior parietal cortex appears most consistently mentioned (*Luria et al., 2016*). Posterior parietal regions have been
associated with the maintenance of online information across time in fMRI studies (*Todd & Marois, 2004*; *Xu & Chun, 2006*). Considering that deficits in neural activity within fronto-parietal systems are among the most consistent findings in ADHD populations (see *Dickstein et al., 2006* for meta-analysis), it is likely that activity in parietal regions, as partly reflected by CDA activity, could play a key role in ADHD pathophysiology related to working memory problems.

To the best of our knowledge, only two studies thus far have investigated neural correlates of storage capacity using the CDA in adults with ADHD and their peers with inconsistent results. *Spronk, Vogel & Jonkman (2013)*, found a smaller CDA for the low load condition in their ADHD sample but generally concluded that there was no evidence for reliable differences in neural filtering efficiency or CDA-$\Delta$ between their ADHD and comparison group. A more recent study by *Wiegand et al. (2016)*, which did not utilize a load manipulation, did find differences in the CDA between individuals with ADHD and controls; showing smaller CDA waves for the ADHD group which also correlated with a patient's symptom severity rating. They concluded that the CDA could be a promising candidate for a neurocognitive endophenotype of adult ADHD.

Given the paucity and conflicting nature of the research in this area to date, we tentatively formulated our hypotheses as follows: (1) the ADHD group, when compared to the comparison group, will show lower CDA-$\Delta$, as operationalized by a difference measure between the CDA of the low and high load condition. We expected this effect to be small since literature has shown visuospatial-WM impairments in ADHD to be neither ubiquitous nor large, especially in college students (*Gray et al., 2016*; *Lambek et al., 2011*; *Nigg et al., 2005*); and (2) the ADHD group will manifest decreased filtering efficiency. We expected this effect to be larger since it may be indicative of a more specific and broader difficulty with selective attention that allows individuals with ADHD to effectively filter out distracting information.

## METHOD

### Sample

Data reported in this manuscript were taken from a larger study investigating changes in neural and behavioral indices after a working memory training program conducted at the University of Toronto from 2011 to 2013 (See also; *Mawjee, Woltering & Tannock, 2015*; *Mawjee et al., 2017*; *Liu et al., 2017*). For the purpose of the current study, 27 unmedicated college students with ADHD were pair-matched with 27 peers on gender (41% female), age (sample mean: 22 years old), and IQ (sample mean: 111). Participants with ADHD were recruited from University Student Disability Services in a major urban area via email lists and flyers. Inclusion criteria were; (1) current enrollment in a post-secondary program, (2) a previous diagnosis of ADHD, (3) registration with respective university or college Student Disability Services, which requires documented evidence of a previously confirmed diagnosis of ADHD (typically, but not invariably in elementary school), and (4) current symptoms consistent with diagnostic criteria for ADHD as indicated by telephone interview and meeting the criterion scores on the 6-item Adult ADHD Self-Report Scale

Part A (ASRS-A). Exclusion criteria were; (1) uncorrected sensory impairment or motor or perceptual handicap that would prevent the use of a computer program, (2) major neurological dysfunction and psychosis or a history of concussion or traumatic brain injury prior to ADHD diagnosis, and (3) limited proficiency in English language. The comparison group was recruited through campus advertisements and they were required to have no history or current presentation of mental health disorders. More details on sample selection can be found in the Supplementary Information.

The study was approved by the institutional research ethics board at the University of Toronto (Protocol reference: #23977). Participants received $20 for their participation. Informed written consent was obtained from all participants prior to beginning the assessments (See copy in Supplementary Information).

## Measures

### Questionnaire and behavioral task measures

Participants completed the Adult ADHD Self-Report Scale (ASRS v1.1; *Adler et al., 2006*) and the Cognitive Failures Questionnaire (CFQ; *Broadbent et al., 1982*; *Wallace, Kass & Stanny, 2002*). The ASRS is a reliable and valid scale for evaluating current symptoms of ADHD and consists of 18 questions based on the criteria used for diagnosing ADHD in the DSM-IV-TR. For inclusion purposes, we used a modified version of the 6-item ASRS screener which was administered by telephone (for more information on this modified version and its psychometric properties, see (*Gray et al., 2014*). The CFQ is a 25-item questionnaire with strong psychometric properties measuring self-reported failures in perception, memory, and motor function in everyday life (See, *Bridger, Johnsen & Brasher, 2013*, for recent evaluation). Four subscales were derived using factor analysis, namely the problems with Memory, Distractibility, Blunder, and (memorizing) Names (See, *Wallace, Kass & Stanny, 2002*).

To assess working memory performance using behavioral (neuropsychological) measures, we administered the Cambridge Neuropsychological Testing Automated Battery (CANTAB). Participants in this task needed to remember the spatial sequence of squares that briefly flashed on the screen in the same order (Forwards subscale) or reverse order (Backwards subscale) they appeared (*Fray, Robbins & Sahakian, 1996*). As an estimate of general intelligence, the Vocabulary and Matrix Reasoning subscales from the Wechsler Abbreviated Scale of Intelligence was assessed (WASI- 2nd Ed; *Wechsler, 1999*).

### Visuospatial working memory EEG task

Our Visual WM task was adapted from *Vogel, McCollough & Machizawa (2005)*. E-prime 1.2 software (Psychology Software Tools Inc., Sharpsburg, PA, USA) was used to control stimulus presentation and timing as well as to collect performance measures. Participants sat in front of a 17-inch VGA monitor at a distance of approximately 80 cm. Our paradigm had several phases, starting with a fixation, a cue, a jitter period, a memory array, retention stage, and a test array, which are all shown in Fig. 1. The fixation cross appeared for 600 ms in the center of the screen after which a cue appeared for 200 ms in the form of an arrow pointing either to the left or right (50% chance) to inform participants as to which side of

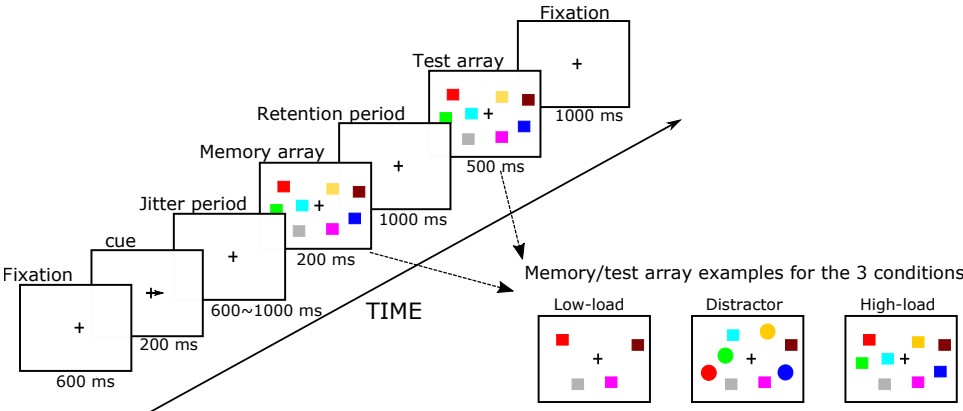

**Figure 1** Simplified flowchart of the visuospatial change detection task with distractors.

their visual field (i.e., left or right hemifield) they should pay attention. After a 400–900 ms jitter period, a memory array appeared of two or four colored shapes in both hemifields for 200 ms in which participants were required to only memorize the colored squares in the cued hemifield. After the memory array disappeared, there was a 1000 ms retention period during which participants needed to maintain the memory array in their WM. Then, the test array appeared and participants were asked to indicate whether or not the test array was identical to the memory array by clicking one of two keys on a keyboard using the index and middle fingers of their dominant hand. On half of the trials, the test array was identical to the memory array, but on the other half, the color of one shape on the attended side was changed. As soon as the response was made, the test array disappeared and the current trial ended. If participants did not make a response within 1,500 ms, the trial ended automatically. The test array was shown for 500 ms after which a fixation appears for 1,000 ms (for task protocol, see Fig. 1).

The task had three conditions: in the low memory load condition (LL), the memory array contained two colored squares in each hemifield; in the high memory load condition (HL), the memory array contained four colored squares. In the distraction condition (DL), the memory array contained two colored squares and two colored circles (see also Fig. 1). Participants were required to only memorize the squares (targets) in the attended hemifield in each condition and ignore the circles (distractors) in the distraction condition. For each condition, these memory items, i.e., squares (size: 0.57° × 0.57°) and circles (diameter: 0.57°), were randomly presented within 5.7° × 8.53° rectangular regions that were centered 2.86° 4 cm to the right and left side of the fixation cross. The color of each item was randomly chosen from 10 easily discriminable colors and a given color only appeared once in the attended hemifield of memory and test arrays. The task had 312 trials, with 104 trials for each condition. Trials of different conditions and of different hemifield-attended sides were randomly mixed and divided within blocks. There were 13 blocks with 24 trials in each block. At the end of each block, participants' accuracy, as well as progress through the task, was shown.

Special care was taken with task instructions considering the nature of our study population. Instructions were presented on several subsequent screens and read aloud by a research assistant who would verify that the participant understood the instructions. For example, participants had to repeat verbally that they were to ignore the circles in the distractor condition. Participants were also told that it was important to keep their eyes focused on the fixation cross in the center of the screen for the duration of the trial and to direct their attention, *and not their eyes*, to the array on the side they were instructed to attend to. Also, they were told to sit in a comfortable, relaxed position and minimize eye blinks, verbalizations, and movements as much as they could. During the practice block, research assistants were actively checking for eye-movements and would correct behavior until participants understood and complied when acting out instructions.

Participants' WM capacity (referred to as 'K') was calculated from their behavioral performance. Similar to *Vogel, McCollough & Machizawa (2005)*, we used the following formula to calculate K: $K = N^{*}(H - F)$, where K is visual WM capacity, N is the number of items in the memory array, H is the hit rate, and F is the false alarm rate. By subtracting F from H, we can correct for guessing and obtain a more accurate capacity value. This value was calculated for the K[HL], K[LL], and K[DL] conditions.

### Neural data acquisition and processing

EEG was recorded with a 128-channel Geodesic Hydrocel Sensor Net at a 500 Hz sampling rate, using EGI Netstation stand-alone software (Electrical Geodesic Inc, Eugene, OR, USA). Data processing was kept similar to previous literature reported on the CDA (e.g., *Vogel, McCollough & Machizawa, 2005*). EGI's Netstation software package was used to filter (.05–30 Hz) and segment the data for correct trials (400 ms before stimulus onset memory array and 1,000 ms post stimulus). Segments containing artifacts were removed using standard automatic algorithms, including rejection of trials contaminated by blinks or eye-movements, and verified manually by a research assistant blind to the study hypotheses (For more details, see, *Liu et al., 2016*). We note that blinks, vertical and horizontal eye-movements were measured by bipolar electrodes placed above and below the left eye and at the outer canthi of both eyes. On average, after removing artifacts, the control group had 71 ($SD = 15$) trials for the High Load, 83 ($SD = 14$) trials for the Low Load, and 80 ($SD = 15$) trials for the Distractor Load. The ADHD group had 62 ($SD = 15$) trials for the High Load, 72 ($SD = 17$) trials for the Low Load, and 66 ($SD = 17$) trials for the Distractor Load. Trial count was not a significant factor when it was added as a covariate in the main analyses.

After average-referencing, data were transferred to Matlab 9.1 (The Mathworks, Inc.) for further processing. Lateral-posterior sites were chosen to calculate the CDA. This selection of electrodes was based on previous studies (e.g., *Woodman & Luck, 1999*) and inspection of the grand average waveform of all subjects. As indexed using the standard EGI system nomenclature, electrode sites 52, 51, 59, 66, 61, 60, and 65 on the left hemisphere and electrode sites 92, 97, 91, 84, 78, 85, and 90 on the right hemisphere were selected (see topoplot figure in the Supplementary Information). First, for each condition and at the 14 electrode sites, ERPs were calculated separately for left and right hemifield-attended trials.

Next, ERPs on the right hemisphere electrode sites were subtracted from the left sites when participants were cued to memorize the right hemifield of memory arrays; and the ERPs on the left hemisphere sites were subtracted from the right sites when participants were cued to the left hemifield. At this point, for each condition and at each electrode site, there was a separate CDA for attended left and attended right hemifield. Then the two sides of the CDA were averaged to produce a single CDA waveform. This procedure was repeated for each of the three conditions at each electrode site. Finally, waveforms from different electrode sites were averaged to produce a final CDA waveform for each condition.

CDA-Δ and filtering efficiency were two measures derived from the CDA waves. CDA-Δ was calculated as the difference, or increases, in amplitude between the CDA amplitudes for the low and high load condition (See also, *Vogel & Machizawa, 2004*). To calculate the filtering efficiency and quantify how efficiently participants can inhibit distractors from entering their limited WM storage, we measured how individuals' CDA amplitudes in the distractors-present condition resembled those CDA amplitudes in the low load (i.e., two items) condition rather than in the high load (i.e., four items) condition. It was assumed that the closer the distractor CDA amplitudes are to those in the low load condition, the higher the filtering efficiency is. Consistent with previous literature (*Vogel, McCollough & Machizawa, 2005*), the following formula was used to calculate this filtering efficiency, $A = (H − D)/(H − L)$, where A is the filtering efficiency, and H, L, and D are the CDA amplitude in high load, low load, and distractor condition, respectively.

## Analytical approach

In general, standard analysis of variance analysis (ANOVA) and $t$-tests were applied to test for differences of Group and Condition. Pearson's $r$ correlation coefficient was used to investigate relationships between behavioral performance, questionnaire, and neural activity. Partial eta-squared values ($\eta^2$) were computed to ascertain effect size. According to *Vacha-Haase & Thompson (2004)*, partial $\eta^2 = .01$ corresponds to a small effect, partial $\eta^2 = .10$ corresponds to a medium effect, and partial $\eta^2 = .25$ represents a large effect.

To test for differences between conditions and groups with the CDA waves, we decided to perform independent $t$-tests along a moving window with 100 ms length and 50 ms overlap for the 300–850 ms duration of the retention period. Within a group, we tested for differences between the High and Low load CDA conditions. Between groups, we tested for differences in filtering efficiency and the CDA-Δ (the CDA-Δ was the H-L difference measure). Previous literature examining the CDA in ADHD reported different time windows for their analysis (e.g., see *Spronk, Vogel & Jonkman, 2013*; *Wiegand et al., 2016*) but we hoped our method would obtain a fine-grained, data-driven, evaluation of temporal differences. The time window in which group differences were maximal would then be used to confirm group differences using an ANOVA.

Computations were conducted with the Matlab Toolbox 'Measures of Effect Size' developed by Harald Hentschk (*Hentschke & Stüttgen, 2011*). To remove the disproportional effect of potential outliers, data were winsorized to 2 standard deviations (see, *Wilcox, 2012*). We set our significance $p$-value to .05 for all analyses. Effects with $p$'s
**Table 1  Descriptive and group differences for questionnaire and behavioral measurements.**

| | | COMP group | | ADHD group | | Group difference statistics | Effect size |
|---|---|---|---|---|---|---|---|
| | | M | SD | M | SD | p | $\eta^2$ |
| ASRS | | 24.48 | 8.80 | 48.25 | 8.39 | <0.001[***] | 0.667 |
| | Total | 31.43 | 8.74 | 58.80 | 13.45 | <0.001[***] | 0.607 |
| | Memory | 4.76 | 2.74 | 12.57 | 5.18 | <0.001[***] | 0.482 |
| CFQ | Distractibility | 14.00 | 4.52 | 25.45 | 5.38 | <0.001[***] | 0.584 |
| | Blunder | 9.38 | 3.23 | 16.19 | 4.49 | <0.001[***] | 0.443 |
| | Names | 4.00 | 2.17 | 5.43 | 2.16 | 0.038 | 0.103 |
| | SSF_RAW | 7.85 | 0.933 | 7.71 | 1.503 | 0.104 | 0.062 |
| | SSF_SS | 0.88 | 0.72 | 0.32 | 1.08 | 0.052 | 0.087 |
| CANTAB | SSB_RAW | 6.75 | 1.48 | 6.65 | 1.82 | 0.849 | <0.001 |
| | SSB_SS | −0.09 | 1.27 | −0.10 | 1.26 | 0.987 | <0.001 |
| | High | 689.84 | 140.66 | 771.12 | 170.15 | 0.166 | 0.057 |
| Reaction Time | Low | 637.44 | 134.62 | 712.90 | 174.08 | 0.189 | 0.052 |
| | Distractor | 673.03 | 135.17 | 735.50 | 170.15 | 0.267 | 0.037 |
| | High | 2.61 | 0.78 | 2.43 | 0.67 | 0.408 | 0.015 |
| CapacityK | Low | 1.80 | 0.17 | 1.72 | 0.22 | 0.152 | 0.044 |
| | Distractor | 1.65 | 0.25 | 1.55 | 0.36 | 0.303 | 0.023 |

**Notes.**
[*] $p < .05$.
[**] $p < .01$.
[***] $p < .001$.

larger than .05 and smaller than .1 were treated as marginal and reported but interpreted only when in predicted directions.

# RESULTS

## Behavioral & questionnaire results

Table 1 shows the means and standard deviations of the questionnaire and behavioral performance tasks of the ADHD group and their peers as well as the statistical tests for group differences. Results confirmed that our ADHD group reported significantly more ADHD symptomatology and everyday cognitive problems compared to their peers. Our ADHD group did not differ from their peers in their visuospatial accuracy as shown by the capacity K measure in our change detection task. Further, no group differences were found for our standardized measure of visuo-spatial WM (all $p$'s > .20).

## Neural results

Figure 2 shows the typical morphology of a CDA waveform for all three task-conditions for the ADHD and comparison group. Statistical tests were ran during the retention period within-group between the high and low load; and between-groups for CDA-Δ and the filtering efficiency. The data showed that in the comparison group, the CDA for the low load was different from the high load condition for the period of 500–900 ms. The ADHD group showed no difference between CDA waveforms in any of the conditions. A one-way
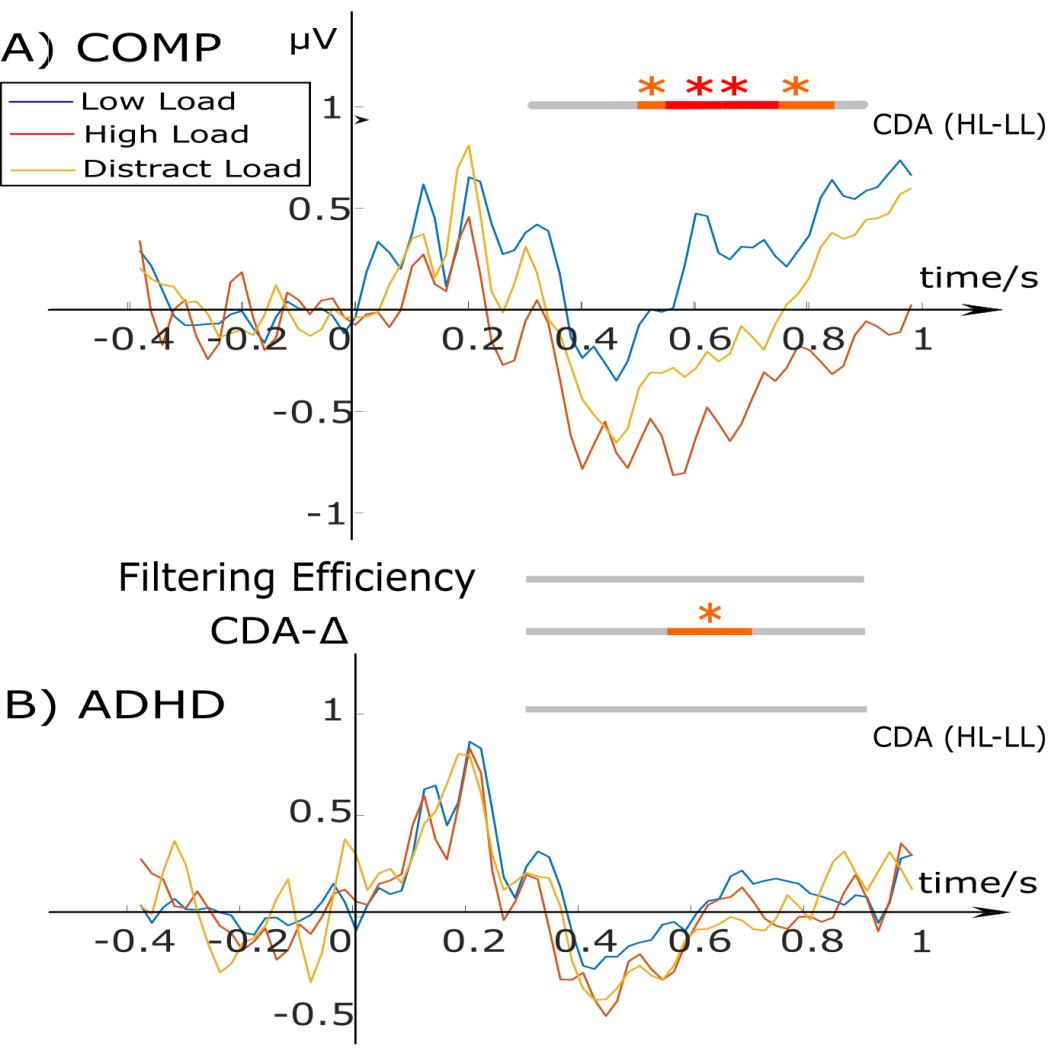

**Figure 2 CDA waveforms for the low load (blue), high load (red), and distractor condition (yellow) for the comparison (A) and ADHD group (B).** Stimulus onset starts at 0 ms for 200 ms. The bars at the top and bottom indicate within-group condition differences between the high (HL) and low load (LL) CDA. The two bars in the middle test group differences for 'CDA-Δ' (difference measure of HL-LL) and 'Filtering Efficiency'. Bars tested differences using a 100 ms time window with 50 ms overlap with the color red representing: **$p < .01$; and orange: *$p < .05$.

ANOVA with Group as a between-subject factor, found a significant group difference for CDA-Δ during the period of 550–700 ms, $F(1, 50) = 5.229$, $p = 0.027$, $\eta^2 = 0.0947$, with the comparison group showing a larger difference between the HL and LL compared to the ADHD group. No statistically significant group-difference effects were found for filtering efficiency at any time point (for that reason, these are not shown in Fig. 2).

### Behavioral and neural results

To investigate the relationship between neural and behavioral effects between different groups, we used the neural data for which group differences were found to be maximal

(e.g., the 550–700 ms window of CDA-Δ). The results were computed separately for the Comparison and the ADHD group. Findings can be summarized by stating that none of the correlations between the neural measures (e.g., CDA-Δ and filtering efficiency) and our questionnaire (e.g., ASRS, CFQ) or behavioral performance measures (e.g., Capacity K and CANTAB) were statistically significant.

## DISCUSSION

The present study investigated whether and how adults with ADHD differ from their peers in aspects of working memory, such as CDA-Δ and filtering efficiency, as differentiated at a neural level using a visual change detection task with distractors. The ADHD group self-reported severe impairments on measures of everyday functioning and ADHD symptomatology. Despite being severely impaired, findings showed that the ADHD group did not differ from their peers on most behavioral indices of working memory performance nor in the neural measure of filtering efficiency. However, the data did reveal a reliable pattern of lower CDA-Δ (e.g., the differentiation between high and low load) for the ADHD group.

The lack of group differences in accuracy measures of our change detection task can, in part, be explained by the nature of our sample: college students with ADHD. These subjects are *relatively* high-functioning adults with ADHD, especially in their performance on standardized neuropsychological tests (See *DuPaul et al., 2009*, for overview). It is possible the task was not sufficiently challenging, particularly in the low load condition as demonstrated by a skewed distribution of accuracy values. Our task, tapping into a basic working memory operation using a simple change detection task, may simply not have been demanding or complex enough to elicit differences at a behavioral level. Previous studies investigating visuospatial working memory in this college student sample using a change detection task with sequentially presented items also failed to find reliable differences in task performance measures (see, *Kim et al., 2014*). Moreover, *Gray et al. (2016)* found that standardized neuropsychological tests may not be ideal in differentiating college students with ADHD from their peers when compared to questionnaire measures, which are likely more sensitive to impairments in relatively unstructured and ongoing myriad of situations found in everyday life (For discussion, see, *Toplak, West & Stanovich, 2013*).

Interpretation of the neural results is less straightforward. We cannot reliably conclude that processes of attentional control (e.g., selective attention) are not involved, as our measure of filtering efficiency requires a separation between CDAs of the high and low load condition. Instead, our data showed perhaps a greater neural impairment in that the brains of individuals with ADHD did not differentiate between load conditions at all. This would suggest they were dedicating a similar amount of neural resources to the high and low load condition. Or perhaps individuals with ADHD have basic problems with encoding stimuli rapidly (*Kim et al., 2014*), which would account for their lack of differentiating low versus high load conditions. Regardless, this pattern can be considered anomalous, and it is possible that such neural processing may explain impairments in visuospatial WM compared to their peers in those unstructured real-life situations as their brains ultimately process such information in a less efficient manner.

Our CDA findings appear to differ from *Spronk, Vogel & Jonkman (2013)* but resemble those of *Wiegand et al. (2016)*. We note that there are number of differences between the studies that could explain discrepant findings. *Spronk, Vogel & Jonkman (2013)* used a smaller sample of adults ($n = 17$) which may not have allowed for the detection of differences at a statistical level and, as *Wiegand et al. (2016)* noted, their task had a maximal set size of three items which may not have been enough to discern load differences optimally.

While it would be premature to characterize the CDA as an endophenotype of ADHD (*Biomarkers Definition Working Group, 2001*; *De Geus, 2010*) and we do caution clinical implications, our study did suggest neural differences in the brain's processing of working memory. Our findings fit in a larger body of literature that has found evidence of anomalous processing at a neural level in adult ADHD of information in working memory with EEG (e.g., *Kim et al., 2014*; *Missonnier et al., 2013*; *Lenartowicz et al., 2014*) and other neural methods (*Ehlis et al., 2008*; *Schweitzer et al., 2000*; *Valera et al., 2005*).

We also want to point out limitations and alternative explanations/potential moderators of our effects. First, the task was not entirely self-paced (i.e., participants had to be continuously engaged and ready for the next trial) and different conditions were presented in a random manner mixed within blocks. Furthermore, there was a jitter period which also made the timing of the onset of the stimulus unpredictable. As such, it is possible the neural effects could also be attributed to difficulties with task switching and readiness or the timed allocation of attentional resources (*Cubillo et al., 2010*; *Dockstader et al., 2008*). Second, an emerging literature is suggesting individuals with ADHD have difficulty with color processing making color a less salient feature, which could also explain our lack of effects in the ADHD group as they were asked to detect color changes (*Banaschewski et al., 2006*; *Kim, Chen & Tannock, 2014*). Finally, the lack of statistically significant correlations between our behavioral and CDA neural findings, in particular for our comparison group, fails to replicate basic findings in the literature. Due to the nature of our population, we were forced to keep trial counts per condition low, or at least lower than they have been in most of the literature in typically developing populations. It is possible that this could have attributed to these non-significant effects. Alternatively, we can also not exclude the possibility that the neural activity we captured during this task may be capturing more general processes involved in the execution, but not reflected in the performance, of the task. The CDA may be a less reliable index of visual WM capacity in individuals with impaired attention and WM functions, such as ADHD patients (*Wiegand et al., 2016*), but also other groups, such as healthy older adults (*Jost et al., 2010*; *Sander, Werkle-Bergner & Lindenberger, 2011*; *Wiegand et al., 2018*).

## CONCLUSIONS

Notwithstanding the aforementioned limitations, the present study contributes to an emerging literature investigating working memory in subjects with ADHD from a neural perspective. Our findings suggest that the brains of individuals with ADHD process the maintenance of information differently from their healthy peers. More research would be needed to confirm our findings and further specify what this means for the characterization of this debilitating condition.

### Funding

This research was supported financially by a CIHR Operating Grant (# 245899, Tannock & Lewis) and by the Canada Research Chair program (RT). Finally, the current project would not have been possible without the support of the College of Education and Human Development at Texas A&M University which provided the first author a scholarship allowing him to pursue this endeavor. There was no additional external or internal funding received for this study. The funders had no role in study design, data collection and analysis, decision to publish, or preparation of the manuscript.

### Grant Disclosures

The following grant information was disclosed by the authors:
CIHR Operating Grant: # 245899.
Canada Research Chair program (RT).
College of Education and Human Development at Texas A&M University.

### Competing Interests

The authors declare there are no competing interests.

### Author Contributions

- Chao Gu analyzed the data, prepared figures and/or tables, authored or reviewed drafts of the paper, approved the final draft.
- Zhong-Xu Liu conceived and designed the experiments, performed the experiments, authored or reviewed drafts of the paper, approved the final draft.
- Rosemary Tannock conceived and designed the experiments, authored or reviewed drafts of the paper, approved the final draft.
- Steven Woltering conceived and designed the experiments, performed the experiments, analyzed the data, prepared figures and/or tables, authored or reviewed drafts of the paper, approved the final draft.

### Human Ethics

The following information was supplied relating to ethical approvals (i.e., approving body and any reference numbers):

The study was approved by the institutional research ethics board at the University of Toronto (Protocol reference: #23977).

### Data Availability

Woltering, Steven; Gu, Chao (2018): Script and Data. figshare. Fileset. https://figshare.com/articles/Script_and_Data/6667499.

Woltering, Steven; Gu, Chao (2018): EEG Raw Data. figshare. Dataset. https://figshare.com/articles/EEG_Raw_Data/6662972.

## Supplemental Information

Supplemental information for this article can be found online at http://dx.doi.org/10.7717/peerj.5601#supplemental-information.

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
