# Peer review of "Neural processing of working memory in adults with ADHD in a visuospatial change detection task with distractors"

_PeerJ, doi:10.7717/peerj.5601_

## Round 0.1 · original submission · Major Revisions

After revisions, the manuscript will be suitable for publication in my own opinion. As raised by both Reviewers, further details are necessary. Moreover, I suggest to discuss in depth the lack of behavioral differences between patients and controls in the task. Finally, it would be useful to find a link with a more limited region in the brain than 'posterior sites'. Could the authors argument on possible brain regions involved in their electrophysiological findings?

·

Basic reporting

This study investigates behavioral (K) and EEG (CDA) correlates of visual working memory in adults with ADHD and a matched control sample. The study is original and employs a well-validated experimental task. The manuscript is generally well-written in good, professional English. The knowledge gap based on existing research on visual working memory in ADHD, research question, and relevance of the study is outlined clearly. The introduction presents the study background briefly, the relevant literature on the topic is introduced and discussed within the present study’s scope. The structure of the manuscript is comprehensible and conforms to the discipline’s standards. Methods are sufficiently detailed and largely sound (see comments below) The figures and the table are relevant, correctly labeled and described, and of good quality. The supplement contains valuable additional information about the broader study context. I could not find any raw data supplied!

Experimental design

Regarding the integration between the behavioral and neural data, I was wondering whether there would be a way to quantify filtering efficiency also in the K measure, for example, differences between K in the presence and absence of distracters? This might be difficult, however, because 2 items will be way under all or most participants’ maximum storage capacity. Perhaps, the relatively low load was also a reason for the lack of group differences in behavior due to ceiling effects?

Generally, the work shows good practice of EEG/ERP methods. However, I suspect the high-pass filter of 0.05 affected the slow waves and reduced the CDA amplitude. This would explain why the CDA amplitude is considerably smaller than in other studies using a similar paradigm (e.g. Vogel & Machizawa, 2004; McColllough et al., 2007). I do not suspect this to change differences between conditions or groups, but effects might be less pronounced. I suggest to repeat the analyses after using a high-pass filter of 0.01 Hz in the pre-processing.

Perhaps related to the issue mentioned above, the CDA waveforms, especially in the ADHD group, seem to not differ significantly from zero in the chosen time window of interest. If the CDA is absent, any effect on the component is difficult to interpret. The method of defining the time window chosen by the authors is also clearly biased by the within-group difference in the control group, that is absent in the ADHD patients and may thus not be optimal to detect any group differences. The maximum negative deflection of the CDA in fact appears to be earlier than the selected time window, around 400. Notably, Wiegand et al., 2016 showed that only the earliest time window of the CDA dissociated between patients and controls and correlated with patients’ symptom ratings.

Validity of the findings

see above, the methodological aspects are also relevant to the validity of the presented results.

Additional comments

In the introduction, a distinction between visual WM and more general WM processes is missing (Eriksson, Vogel, Lansner, Bergström, & Nyberg, 2015). It is also missing a clear explanation (including hypotheses) why the digit span was assessed too, which taps mainly into verbal WM

In the discussion, I believe it is relevant to state that the present study results add evidence to the fact that the CDA is a less reliable index of individual visual working memory capacity in individual’s with impaired attention and visual working memory functions, such as adult ADHD patients (Wiegand et al, 2016), but also other groups, such as healthy older adults (Jost et al., 2011; Sander et al., 2011, Wiegand et al., 2018)

Minor:
Personally, I would include the topography of the CDA in the main manuscript, as the electrode labels described in the methods are not based on the 10-20 system’s nomenclature, which gives the reader information about the location of the selected electrodes.

Note that Wiegand et al. compared individuals with high vs. low VWM capacity, not high vs. low load conditions, as it is stated incorrectly in the introduction

The authors assessed a well-controlled sample that is, as the authors mention, presumably a high-performing subgroup of ADHD patients, who are able to compensate for their deficits to some degree. They are likely to still show symptoms, however, moderately. Do norms/cut-offs of the questionnaires really justify that they were “severely impaired”?

How many participants were excluded based on the stated criteria?

Were subtypes of ADHD classified and considered?

Were the subscales of the CFQ considered?

Reviewer 2 ·

Basic reporting

The work is adequately introduced and and referenced (see general comments for more details).

Experimental design

The work properly designed (see general comments for more details).

Validity of the findings

The validity of the findings needs to be more extensively discussed (see general comments).

Additional comments

In the study “Neural processing of working memory in adults with ADHD in a visuospatial change detection task with distractors” by Gu et al. the authors investigated the neural correlates of working memory deficit in adults ADHD individuals by mean of an even related potential approach (ERP). The authors were interested the correlating the performance in a computerized behavioral task, aimed at measuring both, the efficiency of working memory and attentional filtering, and the corresponding neural activity. The authors also asked if the outcome of a set of neuropsychological related to the pattern of neuronal activation during the working memory task. The authors observed a significant difference between the ADHD individuals and their control group in some of the neuropsychological assessments but a comparable performance in the working memory task. On the contrary, the neuronal pattern of brain activity disguised well the task condition in the comparison group but not in the ADHD group. The authors discuss the difference in neural activity as a dysfunction in visual working memory in ADHD, even though their performance in the task was comparable to that of comparison group.
I found the work well written, adequately introduced and properly designed. In my opinion, the work is suitable for publication.
I have only some comments on the results presentation, interpretation and discussion, especially in light of negative behavioral result.


Results:
Did the authors try to analyze the reaction time in the two groups? Could differences in reaction time among the three conditions be reflected in neural activity, as for instance, a higher value during the retention period predicts a faster response in COMP but not in ADHD?
Also, did the authors test the memory load effect subject by subject? The authors should have enough trial to compare for each participant the performance and the neural activity among the task conditions. If a consistent number of ADHD does not show significant differences between task conditions while the control group does, it should support the author’s hypothesis. Even a smaller difference among the task conditions in ADHD respect to the control group would be consistent with their hypothesis.
I fell confused by the description of the neural results.
The authors describe a between group comparison in the measure of the neural storage capacity. Why did the authors use the term “storage capacity”? They are only comparing different signals. Why should these signals represent a memory capacity if there are no correlations with the behavioral measures?
Which are the signals compared between the one-way ANOVA? If the authors are describing post-hoc comparisons which are the variables compared?
Where in methods this measure is described? Is it the measure A = (H – D)/(H – L) ? If not, where in the paper this measure is used?

Discussion:
The authors should discuss more the fact that the neural activity that they observed is not capturing the processing of memory load, especially in light of a lack of correlation with the behavioral measures in the control group. Looking at the comparable performance between ADHD and the control group in the visual working memory task it seems hard to conclude that they are capturing the ability to process the memory load. It should be rater stressed more the alternative explanations that the neural activity is capturing other characteristics of the task, supporting its execution but not influencing the performance in the task.
There are no mention to the brain areas they are recording from, which role they play in visual-working memory and why they should be dysfunctional in ADHD.
Minor points
Figure2:
P values higher than 0.05 and lower than .1: there is no reason to signal the time point at which these p-values are reached if not discussed in the text. Furthermore, this happens after 800 ms, a time period not considered in other analyses.
Green bars: it is not clear to what statistic each bar is referred to. Does each bar indicate the comparisons Low Load VS High Load, Low Load Vs Distract Load and High Load Vs Distract Load? If so, please use different colors or symbols, otherwise it is not possible to identify at which time each signal is different from another. If, as stated in the legend, the bars indicate the difference between High and Low load I do not see reasons to have three bars.
Red bar indicates the between group statistical difference in which measure? The average signal across the three conditions?

Arrows of x-axis are different in the two panels.
Please indicate in the figure legend or in the figure that the zero of the x-axis is the time of presentation of the array (if I have correctly understood the analysis). Furthermore, it is worth pointing out also when the array disappears (200 ms after the presentation), to indicate to the reader the time point that separates the visual response and the retention time.
Figure legend 2:
Replace: “CDA waveforms for the low load (blue), high load (red), and distractor condition (yellow) for the ADHD (top panel) and Comparison group (bottom panel)” with “CDA waveforms for the low load (blue), high load (red), and distractor condition (yellow) for the ADHD (bottom panel) and Comparison group (top panel)”

---

## Round 0.2 · Minor Revisions

Please address the remaining minor comments

·

Basic reporting

see below

Experimental design

see below

Validity of the findings

see below

Additional comments

The authors addressed all of my previous comments adequately. The manuscript improved further and I believe it is ready for publication. Well done! I only suggest to correct these minor issues in the final version:

l. 14 (abstract) delete “difference in” or “deficits in”

ll. 35-39 it is still not clearly stated that visual working memory is one sub-component of WM, or executive functions more generally. It is also not clear which of the references belong to the one or the other. I’m also not sure any of those show evidence that VWM is “especially” impaired, relative to other processes subsumed under executive functions (e.g. task switching)
l. 332: iconic memory is sensory memory, not working memory (provides WM with the sensory representation to be maintained)

Reviewer 2 ·

Basic reporting

The paper is well written and adequately referenced.

Experimental design

The experimental hypothesis is well introduced.

Validity of the findings

No commenrt

Additional comments

The authors have addressed all raised points. I found the paper improved in the current version and suitable for publication.

---

## Round 0.3 · accepted · Accept

Thank you for responding to comments from reviewers.

#